# Forecasting the July Precipitation over the Middle-Lower Reaches of the Yangtze River with a Flexible Statistical Model

**Qixiao Jiang and Xiangjun Shi ***

School of Atmospheric Sciences, Nanjing University of Information Science and Technology, Nanjing 210044, China
* Correspondence: shixj@nuist.edu.cn

**Abstract:** The multiple regression method is still an important tool for establishing precipitation forecast models with a lead time of one season. This study developed a flexible statistical forecast model for July precipitation over the middle-lower reaches of the Yangtze River (MLYR) based on the prophase winter sea surface temperature (SST). According to the characteristics of observed samples and related theoretical knowledge, some special treatments (i.e., more flexible and better-targeted methods) were introduced in the forecast model. These special treatments include a flexible MLYR domain definition, the extraction of indicative signals from the SST field, artificial samples, and the amplification of abnormal precipitation. Rolling forecast experiments show that the linear correlation between prediction and observation is around 0.5, more than half of the abnormal precipitation years can be successfully predicted, and there is no contradictory prediction of the abnormal years. These results indicate that the flexible statistical forecast model is valuable in real-life applications. Furthermore, sensitivity experiments show that forecast skills without these special treatments are obviously decreased. This suggests that forecast models can benefit from using statistical methods in a more flexible and better-targeted way.

**Keywords:** statistical methods; flexible treatments; the middle-lower reaches of the Yangtze River; precipitation forecast



## 1. Introduction

The middle-lower reaches of the Yangtze River (MLYR) are vulnerable to summer monsoon rainfall [1–5]. There have been many studies about the summer precipitation over the MLYR and how to forecast it [6–11]. Although the theoretical studies have demonstrated that the sea surface temperature (SST) in the preceding winter can provide physically meaningful indicative signals [12–16], its seasonal prediction is still a long-standing challenge for the operational forecasting community [17,18]. The methods for its prediction with a lead time of one season are usually classified into two distinct categories: statistical forecast models and numerical forecast models [18,19]. Many previous studies about the summer precipitation over South China indicated that the statistical forecast model is more skillful [11,20], and summer prediction may be beneficial if the summer is divided into different time periods [21–23]. In this study, a linear statistical forecast model for July precipitation over the MLYR based on the latest winter SST was developed. In order to improve its forecasting skills, some special treatments (i.e., more flexible and better-targeted methods) were introduced in the forecast model. Furthermore, considering the actual demand of the operational forecasting community, more attention was paid to the abnormal years (i.e., years with an obviously higher or lower precipitation) in developing this forecast model.

Unlike previous studies which focus on introducing reasonable predictors used for the linear statistical forecast model (e.g., [24,25]), this study focuses on how statistical analysis methods can be fully exploited based on the characteristics of observed samples and climate background knowledge. Some special treatments (i.e., more flexible and

better-targeted methods) were used in the forecast system. Firstly, the MLYR domain in this study was slightly different from previous studies. The boundary of the MLYR was redefined based on the rain gauge stations with similar predictors. Secondly, in order to make full use of the forecasting information from the prophase winter SST, the appropriate predictors (i.e., SST variables) used for establishing the forecast model were dynamically calculated based on the observed samples (from 1951 to the year before the forecasting year). This is different from traditional statistical forecast models with a fixed number of predictors. Thirdly, in order to take advantage of the previous theoretical knowledge, some artificial samples derived from the theoretical knowledge (hereafter referred to as theoretical samples) were added to the observed samples. In other words, the theoretical knowledge about the relationship between input predictors (i.e., the prophase winter SST) and target predictands (i.e., July precipitation over the MLYR) was used to somewhat constrain the forecast model. Finally, the samples from abnormal years were amplified in establishing the forecast model because abnormal years usually provide stronger indicative signals as compared to normal years [12,26]. Besides the above special treatments, how the forecast skills of the statistical model can be estimated was also carefully considered. To avoid the over-fitting problem, forecast skills were calculated from the rolling forecast experiment. At last, the interpretability of the forecast model was also investigated.

In this study, the linear statistical forecast model for July precipitation over the MLYR was introduced first, and the effects of the above-mentioned special treatments on the forecast skills were then analyzed. The paper is organized as follows: Section 2 introduces the forecast system and required input data, the performance analysis is given in Section 3, and Section 4 provides a discussion about the successful experience of developing a linear statistical forecast model. Conclusions are provided in Section 5.

## 2. Data and Methods

### 2.1. The Input Data

Monthly precipitation data from 160 stations in China from 1951 to 2021, provided by the National Climate Center of the China Meteorological Administration, were used in this study [27,28]. The July precipitation anomaly percentage over the MLYR (i.e., the target predictand) was calculated based on the rain gauge stations within the MLYR domain. In order for all rain gauge stations in the MLYR to have similar indicative signals, the boundary of the MLYR domain was tuned based on the composite analysis of each station. Figure 1 shows the MLYR domain used in this study and the composites of the prophase winter SST anomalies in low and high precipitation years. As compared to the traditional geographical MLYR domain from previous studies (e.g., [25]: 28° N to 32° N, 110° E to 122° E), the MLYR domain defined in this study (hereafter referred to as the predictor-based MLYR domain) includes a few western stations (e.g., Station A) and excludes a few eastern stations (e.g., Station C and D). The composites of the prophase winter SST anomalies at Station B are in agreement with previous theoretical studies that the positive anomalies of the winter SST in the equatorial central and eastern Pacific might result in a positive precipitation anomaly over the MLYR [29,30]. Station A has similar composites to Station B. This is the reason why Station A is included within the predictor-based MLYR domain. Besides Station A, the other stations marked with red dots also have similar composites to Station B (not shown). Stations C and D are excluded from the predictor-based MLYR domain because their composites are obviously different from Station B. In this study, years with precipitation anomaly percentages lower than $-25\%$ or higher than $25\%$ are considered as abnormal years. From 1951 to 2021, the numbers of high and low abnormal years are 15 and 20, respectively. In order to enhance the contribution of abnormal years in establishing the forecast model, the precipitation anomaly percentages of abnormal years were amplified. Table 1 lists the original percentage and corresponding amplified percentage.

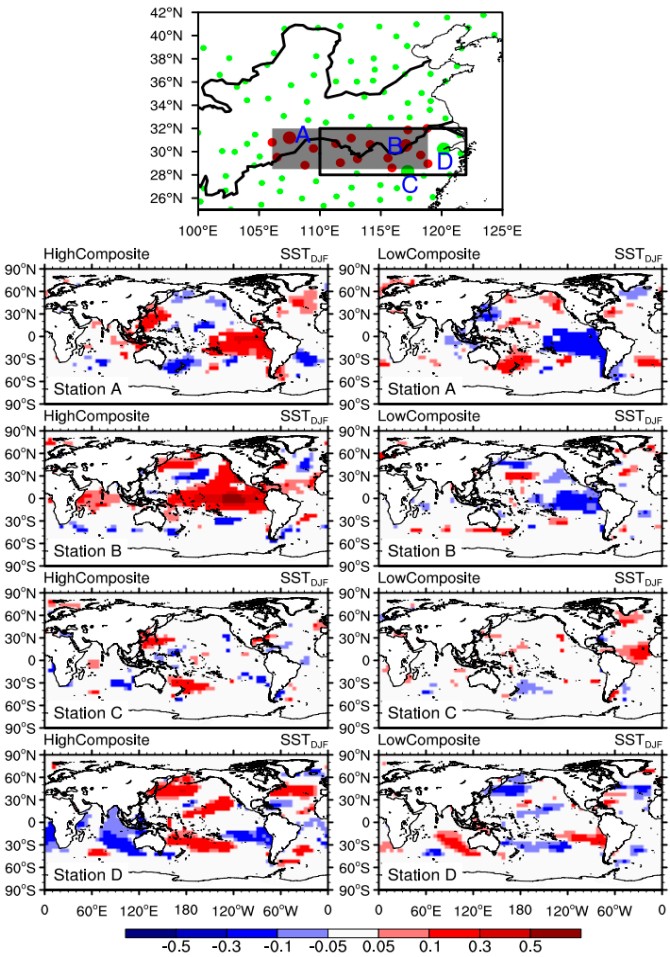

**Figure 1.** The rain gauge stations (red dots in the grey shaded area) are selected for calculating the precipitation over the MLYR (**upper panel**). The rectangular borderlines indicate the traditional geographical MLYR domain. The lower panel shows the composites of the prophase winter SST anomalies in high (**left column**) and low (**right column**) precipitation years for the four rain gauge stations marked in the upper panel. The station labels (i.e., A, B, C, and D) are shown at the lower left corner.

**Table 1.** The original precipitation anomaly percentage (A, %) and corresponding amplified anomaly percentage (Â, %).

| Original Anomaly Percentage (A, %) | Amplified Anomaly Percentage (Â, %) |
|:---:|:---:|
| A < −25% | Â = A − 25% |
| −25% < A < 25% | Â = A |
| A > 25% | Â = A + 25% |

The SST data (i.e., the raw input predictors) were obtained from Physical Sciences Laboratory. This dataset is stored on a $5° \times 5°$ grid and consists of monthly anomalies from 1856 to the present [31]. Unlike the precipitation over the MLYR, the SST shows an obvious decadal-scale warming trend due to global warming (not shown). This decadal-scale trend is removed by the 21-point (i.e., year) moving average method. Correspondingly, the precipitation anomaly percentage is calculated based on the 21-year moving average. Because the SST indicative signals are mostly located in the Pacific and Indian Ocean (Figure 1), only the Pacific and Indian Ocean (40° S to 60° N, 30° E to 90° W) winter-average SST anomalies were used in the forecast model. The previous theoretical studies usually analyze the impact of one SST variable (e.g., the index of Nino3.4) on precipitation

(e.g., [32,33]). It is clear that only one SST variable cannot extract all the indicative signals provided by the SST field. On the other hand, if the SST grid variables were used as input predictors, a high number of input variables (>1000) would be redundant because the SSTs between adjacent grids are very similar. In this study, the 20 leading empirical orthogonal function (EOF) eigenvectors of observed winter SST anomalies are stored as prescribed fixed patterns. The corresponding 20 leading SST principal components (i.e., 20 SST variables) are used as candidate predictors. Among them, the contribution rate of the first principal component is 40.7%, which is much higher than the contribution rate of other principal components.

Theoretical relation (i.e., physical mechanism) has much greater reliability than statistical relation [34,35]. Previous studies have discovered many physical mechanisms for the impact of the prophase winter SST anomalies on summer precipitation [36,37]. For instance, it has been indicated that [38] the warm winter SST anomalies in the equatorial central and the eastern Pacific would enhance the summer precipitation over the MLYR and illustrate the corresponding physical mechanisms. In order to incorporate the theoretical knowledge into the statistical forecast model, we artificially produced some samples based on the theoretical knowledge (i.e., theoretical samples). Figure 2 shows the composites in the high and low precipitation years. Here, only the abnormal years consistent with the above theoretical studies were selected for this analysis. The low composite shows a general opposite pattern to the high composite, although there are some obvious differences. Furthermore, it is reasonable to assume that the precipitation would be weakened/enhanced if the composited prophase SST anomaly was weakened/enhanced [32,38]. Based on this background knowledge, 20 theoretical samples were produced by reducing or enhancing the high composite data. For example, both the composited SST field (Figure 2, left) and corresponding precipitation anomaly percentage (i.e., 81%) multiplied by 1.12 (or other values around 1.0) can produce a theoretical sample. Similarly, another 20 theoretical samples were produced based on the low composite data (Figure 2, right). A total of 40 theoretical samples were produced and stored as prescribed data. These prescribed theoretical samples would be used together with observed samples. The number of theoretical samples decides the contribution of theoretical relation to the forecast model. If the number is very small, the contribution is small. If the number is very large, the indicative signal from observed samples becomes negligible. Sensitivity experiments with three different theoretical sample numbers (20, 40, and 80) showed that the performance with 40 is slightly better than that with the other two (not shown). Note that the high/low precipitation from the composite analysis (Figure 2) is not only affected by the corresponding composited winter SST field but also by other factors (including uncertain factors). As a result, these theoretical samples amplify the contribution of the prophase SST to the corresponding precipitation.

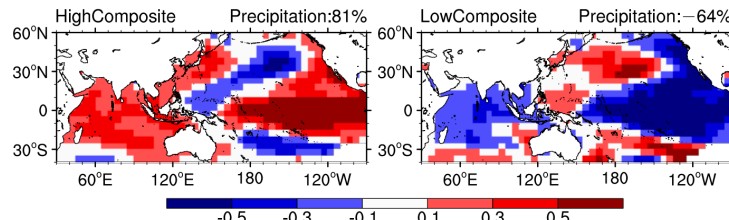

**Figure 2.** The composites of the prophase winter SST anomalies in high and low precipitation years (precipitation over the MLYR). The corresponding high and low precipitation anomaly percentages are shown at the top right corner.

## 2.2. Forecast Model and Experiments

In this study, the forecast model was established by the multiple linear regression equation between precipitation and SST predictors (i.e., SST principal components). As the number of predictors increases, the regression equation can fit the given samples ever well, which causes an over-fitting problem [39]. Therefore, choosing the appropriate

predictors is a very important step. Here, two approaches were used to determine the appropriate predictors (the number of predictors was much lower than 20): stepwise regression (RegStep) and leave-one-out regression (RegLOO). The stepwise regression procedure consists of iteratively adding and removing predictors (i.e., stepwise selection). At last, the statistically significant predictors are chosen to establish the final regression equation [40,41]. To improve efficiency, the order of using these 20 SST candidate predictors was determined by their Pearson's linear correlation coefficients (Cor) with precipitation. In the RegLOO, first, the leave-one-out cross-validation approach [42,43] was used to quantify the forecast skill of the multiple linear regression equation with different combinations of predictors. Afterwards, the best combination of predictors was found based on the forecast skills. As suggested by the name of leave-one-out cross-validation, one year (i.e., one subset) is chosen as the validation set, and the remaining years are used for establishing the regression equation. This continues until all years have acted as the validation set. For instance, in the RegLOO with a given combination of predictors, all observed samples (e.g., 1951~2020, assuming the current time is 1 March 2021) were divided into 70 subsets, each containing one sample (i.e., one year). Firstly, the regression equation was established based on the 2nd to 70th samples (i.e., 1952~2020), and the first prediction (i.e., the predicted 1951 precipitation) was calculated with this regression equation and the SST variables from the 1st sample (i.e., 1951). Subsequently, the regression equation was established again based on the 1st and the 3rd to 70th samples (i.e., 1951 and 1953~2020), and the second prediction (i.e., the predicted 1952 precipitation) was calculated with this new regression equation and the SST variables from the 2nd sample (i.e., 1952). This procedure was repeated until the 70th prediction (i.e., the predicted 2020 precipitation) was calculated. Finally, the forecast skill using this given combination of predictors was evaluated by the Cor (correlation) between the predictions from 1951 to 2020 and corresponding observations. So far, one given combination of predictors corresponds to one forecast skill (i.e., correlation). Here, 20 combinations are tested. The first combination only uses one SST predictor, which has the strongest relation with precipitation. The second combination uses two SST predictors, which are the first two predictors based on correlation sorting. This continues, and the best number of predictors can be found based on corresponding forecast skills. After the appropriate predictors are fixed, the regression equation can be established again with all observed samples. Leave-one-out cross-validation is one of the methods most widely used to estimate the forecast skills, which somewhat excludes the over-fitting problem. More details about leave-one-out regression can be found in some textbooks about statistical methods (e.g., [40]).

In this study, the forecast skills were estimated by rolling forecast experiments. One rolling forecast experiment consists of two parts: forward rolling forecast experiment (from 1986 to 2021) and backward rolling forecast experiment (from 1985 to 1951). In the forward rolling forecast experiment, all available samples (from 1951 to the year before the forecasting year) were used for establishing regression equation. Assuming it was 1 March 1986, the July 1986 precipitation was predicted using the regression equation established by the observed samples from 1951 to 1985. Analogously, this was done for subsequent years. The backward rolling forecast experiment was carried out under the assumption that the order of years is reversed. Besides the Cor, the forecast skills were also estimated by the ratio of successfully predicted abnormal years (Succ) and the ratio of contradictory predictions of the abnormal years (Bad). Succ is the number of successfully predicted abnormal years divided by the number of observed abnormal years. Bad is the number of opposite predicted abnormal years (i.e., the prediction is high/low, but the observation is low/high) divided by the total number of predicted abnormal years. Because both theoretical samples and the amplification of abnormal precipitation enhance the contribution of SST to precipitation, the predicted precipitation is compressed by multiplying it by a parameter. In order for the number of predicted abnormal years to be close to the observations, the parameter was set to 5/6 in this study. In order to test the effects of special treatments, four sensitive experiments were performed. Table 2 lists

all the experiments carried out in this study. The REF experiment includes all the special treatments introduced above. The MLYR domain used in the G-DOMAIN experiment was defined by a traditional geographical concept. The difference between the REF and G-DOMAIN experiments illustrates the advantage of using the predictor-based domain. Theoretical studies commonly focus on the impact of one or two SST variables (e.g., the first principal component) on precipitation. In order to investigate how much information one or two SST variables can extract from the SST field, only one/two principal components were used for establishing the regression equation in the FEWER-P experiment. In order to test the effect of theoretical samples, the NO-T-SAMPLE experiment was carried out without theoretical samples. As compared to the REF experiment, original precipitation anomaly percentages were used in the NO-AMPLIFY experiment. This experiment was used to test whether amplifying abnormal samples can improve forecast skills.

**Table 2.** List of all the forecast experiments.

| Experiments | Description |
| --- | --- |
| REF | Reference rolling forecast experiment. This experiment was run twice using RegStep and RegLOO, respectively. |
| Sensitivity experiments for the special treatments | |
| G-DOMAIN | Similar to REF but the MLYR domain was defined by traditional geographical concept (Figure 1). |
| FEWER-P | Similar to REF but only one/two SST principal components were used for establishing the regression equation. |
| NO-T-SAMPLE | Similar to REF but without theoretical samples. |
| NO-AMPLIFY | Similar to REF but the precipitations from abnormal years were not amplified. |

## 3. Results

### 3.1. Performance and Forecast Skill

During the rolling forecast process, the best combination of predictors used for establishing the regression equation might change for different years. In the REF experiment with the RegStep approach, the number of predictors (i.e., the SST principal components) used for the final regression equation was in the range of 4~8, which is no more than 10% of the corresponding total number of samples (75~110, including theoretical samples). Similar to the RegStep approach, the number of predictors determined by the RegLOO approach was also in the range of 4~8, except in a few years. Furthermore, for the same forecast year, the predictors selected by the LOO approach were generally similar to those selected by the RegStep approach. It is noteworthy that the RegStep approach has a much shorter processing time than the RegLOO approach because the RegLOO approach needs to calculate forecast skills with 20 different predictor combinations.

Figure 3 shows the results of the REF experiment. The Cor with the RegStep and RegLOO approaches were 0.464 and 0.504, respectively. From 1951 to 2021, the number of observed abnormal years (both high and low) was 35. The numbers of successfully predicted abnormal years with RegStep and RegLOO approaches were 19 and 20, respectively. The Succ from these experiments (RegStep19/35 and RegLOO 20/35) indicates that more than half of the observed abnormal years were successfully predicted. In terms of high abnormal years, the number of observed abnormal years was 15. Both RegStep and RegLOO yielded nine successfully predicted high abnormal years. Three-fifths of the observed high abnormal years were successfully predicted. The numbers of predicted abnormal years with RegStep and RegLOO approaches were 29 and 34, respectively. The Bad from these experiments (RegStep 0/29 and RegLOO 0/34) shows that neither RegStep nor RegLOO yielded a contradictory prediction of the abnormal years. Unsuccessful predictions in some years (e.g., 1954) are still unavoidable. It is difficult for statistical models to improve the predictability of these uncommon years (e.g., 1954) because statistical models only catch common characters. More discussion about 1954 is provided in Section 3.1. At last, it is

necessary to point out that the forecast skills with the RegStep and RegLOO approaches are similar because the predictors selected by the RegStep and RegLOO approaches are generally similar.

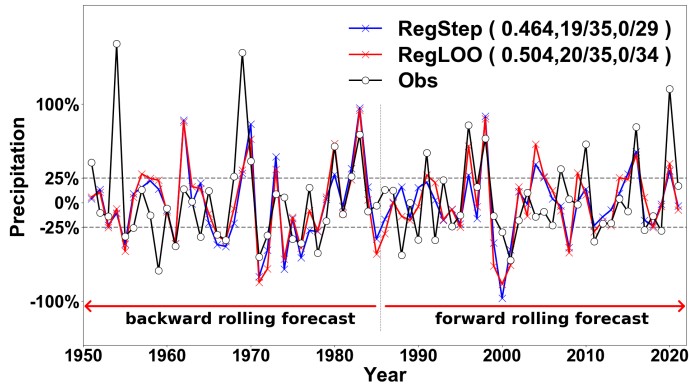

**Figure 3.** The precipitation anomaly percentage (%) from observations (Obs, black line) and rolling forecast experiments with the RegStep (blue line) and RegLOO (red line) approaches. The dashed lines indicate the precipitation thresholds for abnormal years. Forecast skills (Cor, Succ, and Bad) are shown after the corresponding experiment names.

### 3.2. Effects of Special Treatments

This section illustrates the effects of the special treatments in the forecast model. To facilitate comparative analysis, Table 3 list the forecast skills (Cor, Succ, and Bad) from all experiments.

**Table 3.** List of the forecast skills (Cor, Succ, and Bad) from all experiments.

| Experiments | | Cor | Succ | Bad |
|---|---|---|---|---|
| REF | RegStep | 0.464 | 19/35 | 0/29 |
| | RegLOO | 0.504 | 20/35 | 0/34 |
| G-DOMAIN | RegStep | 0.445 | 18/37 | 2/34 |
| | RegLOO | 0.393 | 16/37 | 2/37 |
| FEWER-P | Reg1eof | 0.294 | 12/35 | 3/26 |
| | Reg2eof | 0.402 | 13/35 | 2/26 |
| NO-T-SAMPLE | RegStep | 0.374 | 11/35 | 1/15 |
| | RegLOO | 0.362 | 13/35 | 1/23 |
| NO-AMPLIFY | RegStep | 0.460 | 15/35 | 1/22 |
| | RegLOO | 0.425 | 15/35 | 1/25 |

After using the traditional geographical concept of the MLYR domain (i.e., the G-DOMAIN experiment), the average precipitation over the MLYR was different from that over the predictor-based MLYR domain. The number of observed abnormal years increased to 37, and the theoretical samples were also recreated. As expected, the forecast skills (Cor, Succ, and Bad) from the G-DOMAIN experiment were decreased as compared with the REF experiment (Figures 3 and 4a). The Cor from the G-DOMAIN experiment with RegStep and RegLOO approaches decreased to 0.445 and 0.393, respectively. Both RegStep and RegLOO yielded two contradictory predictions of the abnormal years. The successfully predicted abnormal years did not reach 50% (RegStep 18/37 and RegLOO 16/37). The decline with the RegStep approach was markedly less dramatic than that with the RegLOO approach. One possible reason for this is that the RegStep approach is more robust than the RegLOO approach in extracting appropriate predictors.

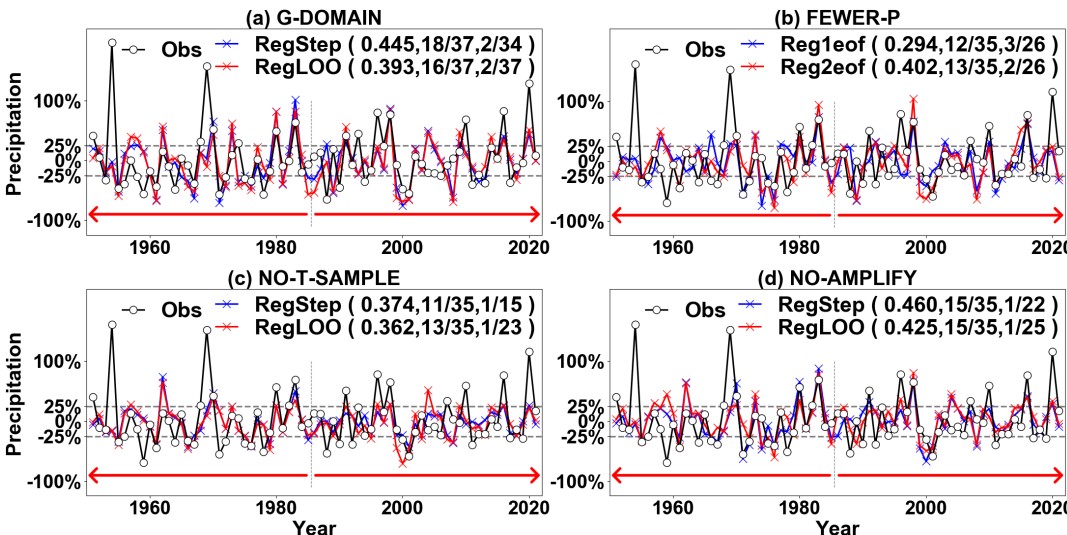

**Figure 4.** Similar to Figure 3, but for sensitivity experiments. The experiment name is shown at the top of each graph.

The FEWER-P experiment was run twice using one SST principal component (i.e., the first SST principal component, Reg1eof) and two SST principal components (most related to precipitation, Reg2eof), respectively. As compared with the REF experiment, the correlations from the FEWER-P experiment (Figure 4b; Reg1eof 0.294 and Reg2eof 0.402) were decreased due to the lower number of predictors. Meanwhile, the variances of predicted precipitations were also decreased. The standard deviations from the Reg1eof and Reg2eof experiments were 0.30 and 0.34, respectively. These are lower than those from the REF experiment (RegStep 0.36 and RegLOO 0.38). As a result, the number of predicted abnormal years decreased to 26 in both the Reg1eof and Reg2eof experiments. The number of successfully predicted abnormal years was also obviously decreased (Reg1eof 12/35 and Reg2eof 13/35). Furthermore, there were two or three contradictory predictions of the abnormal years. In short, the forecast skills are obviously decreased if only one or two SST principal components are used. This indicates that one or two SST variables are not enough to catch the potential indicative signals from the entire SST field.

The theoretical samples not only constrain the forecast model (i.e., the regression equation) by composite analysis, but also amplify the contribution of the composited SST to the corresponding precipitation. Without these theoretical samples, the predicted precipitations become weaker, especially for the years with a prophase SST field similar to the composited SST in high/low precipitation years (Figure 4c). For example, the prophase SST field of 1998 was similar to the composited SST in high precipitation years. The predicted precipitation anomaly percentages from the REF experiment with the RegStep and RegLOO approaches were 87.6% and 85.0%, respectively. In the NO-T-SAMPLE experiment, the predicted precipitation anomaly percentages of 1998 were obviously decreased (RegStep 28.4% and RegLOO 28.9%). The numbers of predicted abnormal years with the RegStep and RegLOO approaches decreased to 15 and 23, respectively. The Succ from the NO-T-SAMPLE experiment reduced to about one-third (RegStep 11/35 and RegLOO 13/35). Both RegStep and RegLOO yielded one contradictory prediction of the abnormal years. Meanwhile, the correlations from the NO-T-SAMPLE experiment were also obviously decreased (RegStep 0.374 and RegLOO 0.362). The comparisons between the NO-T-SAMPLE and REF experiments suggest that theoretical samples are useful for improving forecast skills.

Figure 4d shows the results of the NO-AMPLIFY experiment. As expected, without amplifying abnormal precipitation, the numbers of predicted abnormal years were obviously decreased (RegStep 22 and RegLOO 25). The Succ were also obviously decreased (RegStep 15/35 and RegLOO 15/35) as compared with the REF experiment. Furthermore, the correlations from the NO-AMPLIFY experiment were also decreased (RegStep 0.460 and

RegLOO 0.425), and both RegStep and RegLOO yielded one contradictory prediction of the abnormal years. The decreases in Bad and Cor in the NO-AMPLIFY experiment suggest that enhancing the contribution of abnormal years to establishing regression equations might be helpful for improving forecast skills.

*3.3. Interpretability of the Forecast System*

Although the forecast skills from the rolling forecast experiments are acceptable, it is still necessary to check the interpretability of the forecast model. However, it is difficult to illustrate the physical mechanism of a complex statistical forecast model (e.g., multiple regression with several predictors). Fortunately, it is easy for statistical forecast models to show the relations between input predictors and output predictands, which can be calculated by the perturbation-based method.

In this study, the input–output relations from the regression equations used in the REF and Reg1eof experiments were investigated by the sensitivities of precipitation to grid SST anomalies (Figure 5). The grid value indicates the change in precipitation (in units of %) caused by increasing SST by 1 K in the grid. Because the SST values between adjacent grids are similar, the sensitivity field must be analyzed on a much larger scale. The inner product of the prophase SST anomaly field and sensitivity field determines the corresponding year's precipitation. In other words, the sensitivity field intuitively represents the corresponding regression equation used in the rolling forecast experiment. Because only the first SST principal component was used in the Reg1eof experiment, the sensitivity fields from the Reg1eof experiment are equal to the first EOF eigenvector multiplied by the corresponding regression coefficient. The patterns of these sensitivity fields are similar to the high/low composition analysis (Figure 2). This is also the main reason why the forecast skills of the Reg1eof experiment were not too bad. As compared to the Reg1eof experiment, the sensitivity fields from the two REF experiments (i.e., RegStep and RegLOO) became complex and intense because several SST principal components were selected in the regression equations (Figure 5). The sensitivity field from the REF experiment with the RegStep approach was very similar to that with the RegLOO approach. Furthermore, under the same rolling forecast experiment, sensitivity fields among different years (not only the four years shown in Figure 5, not shown) were also similar. This suggests that the linear statistical forecast model is relatively stable among different years.

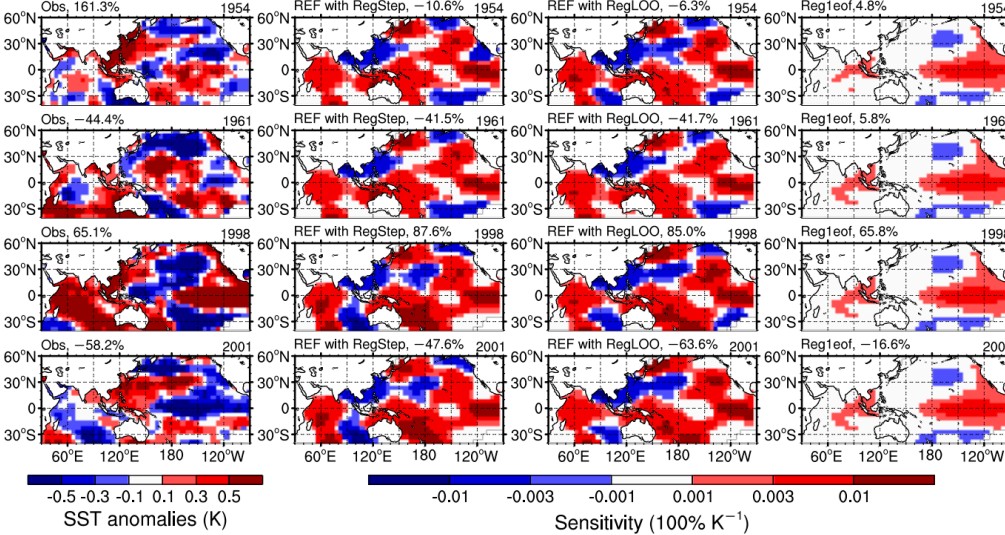

**Figure 5.** The prophase SST anomalies in four years (**left panel**) and the forecast model sensitivities of the corresponding year (**right panel**). The experiment name, precipitation percentage anomaly, and year are shown at the top of each graph.

This paragraph analyzes these sensitivity fields from the view of improving forecast skills. The prophase SST anomaly field from 1998 (high precipitation year, the third row in Figure 5) is similar to the high composite analysis (Figure 2, left). As a result, Reg1eof and two REF experiments yielded a successful prediction. The year 2001 was a low precipitation year (−58.2%). Although the prophase SST anomaly field from 2001 (the fourth row in Figure 5) was similar to the low composite analysis (Figure 2, right), the predicted precipitation from the Reg1eof experiment was not low enough (−16.6%). The main reasons can be found through the comparison between 2001 and 1998. The prophase SST anomaly field from 2001 was weaker than that from 1998, and the pattern of the 2001 prophase SST anomaly was not opposite to the sensitivity field in a few regions (e.g., South China Sea). Note that the predicted precipitations from two REF experiments were low enough (−47.6% and −63.6%). This indicates that the more complex sensitivity fields (i.e., regression equation) from two REF experiments could better fit the observations. The low precipitation year 1961 (−44.4%) is another example. Because the prophase SST anomaly field from 1961 (the second row in Figure 5) showed positive anomalies in the central Pacific (i.e., one characteristic of high precipitation years), the predicted precipitations from the Reg1eof experiment failed (a positive value of 5.8%). However, the predicted precipitations from two REF experiments were successful (−47.6% and −63.6%). At last, unsuccessful predictions in some years (e.g., 1954) are unavoidable for statistical models. The year 1954 was uncommon. The precipitation anomaly percentage was very high (161.3%), although the prophase SST anomaly field (the first row in Figure 5) was obviously different from the high composite analysis (Figure 2, left). The predictions from two REF experiments failed because the stable sensitivity fields (i.e., regression equations) could only fit most years (not the uncommon years). Taken overall, in order to better fit the observations, the input–output relations from the multiple regression equations become complex and intense as compared with single-factor regression equations.

## 4. Discussion

It is useful to discuss the experience of developing a meteorological/climatological forecast model with statistical methods. Whether complex machine-learning methods or simple linear statistical methods are adopted, the statistical methods can be applied flexibly according to the characteristics of the statistical objects. On the other hand, it is necessary to check the interpretability of the forecast model via the input–output relations (which is feasible).

Generally speaking, statistical methods in mathematics must have a high generalization ability. In other words, these statistical methods cannot directly take full advantage of the specific characteristics of statistical analysis objects. Unlike other application fields, there has been a lot of theoretical background knowledge in meteorology and climatology. The specific characteristics of analysis objects should be considered when developing meteorological/climatological forecast models. Therefore, in this study, some special treatments (e.g., the domain of the input SST field and the theoretical samples produced based on composite analysis) were carried out before the regression equation was established. Sensitivity experiments suggest that these special treatments are useful for improving forecast skills. The theoretical background knowledge can not only be used to select physically based predictors, but also has more applications.

Theoretical knowledge is usually developed based on statistical analysis. In other words, statistical analysis usually precedes theoretical knowledge. Therefore, there seems to be no fixed standard on how to judge the interpretability of statistical forecast models. The input–output relations from forecast models can precede the current theoretical knowledge. In this study, the sensitivity fields from the single-factor (i.e., the first SST principal component) regression equations were well in agreement with the current theoretical knowledge (i.e., the composite analysis). The more complex sensitivity fields from multiple regression equations, which can yield better predictions, indicate that most regions have an obvious impact on precipitation, and the total impact of the input SST field on precipitation is a

result of the contributions of every grid. In the future, this might be confirmed by physically based numerical simulation studies that investigate the impact of the input SST field as a whole.

## 5. Conclusions

In this study, a linear statistical forecast system for July precipitation over the MLYR was developed. The latest winter SST anomaly field (40° S to 60° N, 30° E to 90° W) rather than one/two SST variables was used as input predictors. Considering the actual demand, more attention was paid to the abnormal years. Special treatments were added to this forecast system according to the characteristics of the observed samples and related theoretical knowledge. The main purpose of this study was to investigate the effects of more flexible and better-targeted statistical methods (i.e., special treatments) on forecast skills.

The REF rolling forecast experiments (both RegStep and RegLOO) show that the Cor between prediction and observation are around 0.5. Furthermore, the forecast skills of the abnormal years (i.e., Succ and Bad) show that more than half of the abnormal years were successfully predicted, and there was no contradictory prediction of the abnormal years. Furthermore, sensitivity experiments indicate that the targeted special treatments used in the forecast model are helpful for improving forecast skills. These special treatments include a flexible domain definition, the extraction of indicative signals from SST field, theoretical samples, and the amplification of abnormal precipitations. Among them, the successful experience of theoretical samples shows high generality and flexibility. This experience can also be used for unilinear statistical forecast models (e.g., machine learning models) insofar as there is theoretical knowledge of the relation between input predictors and output predictands. The extraction of indicative signals from the prophase SST field is an important step. Here, 20 leading SST principal components (i.e., 20 variables calculated by the projection onto given eigenvectors) were used as candidate predictors. After that, the appropriate predictors were selected objectively by two approaches (RegStep and RegLOO), respectively. The differences in forecast skills between RegStep and RegLOO are not obvious. The RegStep approach is preferred because it has a much higher computational efficiency. The reason why forecast models with several SST variables (i.e., principal components) can yield better predictions than those with only one SST variable is illustrated by interpretability analysis (i.e., the input–output relations from forecast models). Besides the method of extracting SST indicative signals introduced in this study, there might be other methods worth investigating. Taken overall, it is necessary for forecast models to use statistical methods in a more flexible and better-targeted way according to the characteristics of observed samples and related theoretical knowledge.

**Author Contributions:** X.S. designed this study, wrote the Fortran code of the forecast model, and carried out the experiments used in this study. Q.J. created all figures. X.S. and Q.J. wrote the manuscript. All authors have read and agreed to the published version of the manuscript.

**Funding:** This research was funded by the National Key Research and Development Program of China (grant nos. 2018YFC1507001) and the National Natural Science Foundation of China (grant nos. 41775095 and 42075145). The APC was funded by the same funders.

**Institutional Review Board Statement:** Not applicable.

**Informed Consent Statement:** Not applicable.

**Data Availability Statement:** The precipitation data are downloadable from the National Climate Center of the China Meteorological Administration (http://cmdp.ncc-cma.net/cn/index.htm accessed on 6 December 2022). The original SST data are downloadable from https://psl.noaa.gov/data/gridded/data.kaplan_sst.html (accessed on 6 December 2022). The code of the forecast system and related results used in this study have been archived in a public repository https://doi.org/10.5281/zenodo.7405010 (accessed on 6 December 2022).

**Acknowledgments:** This study was conducted in the High-Performance Computing Center of Nanjing University of Information Science & Technology.

**Conflicts of Interest:** The authors declare no conflict of interest.

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
