# Peer review of "Forecasting the July Precipitation over the Middle-Lower Reaches of the Yangtze River with a Flexible Statistical Model"

_atmosphere, doi:10.3390/atmos14010152_

Round 1
Reviewer 1 Report
1. L84-86. The MLYR domain defined in this study includes a few western stations ( Station A and other stations marked with red dots ) and excludes a few eastern stations (e.g., Station C and D). It is recommended that all stations not included in rectangle are marked in green.
2. What is the basis for the value of the corresponding amplified anomaly percentage in Table 1?
3. L149-151. How to get the optimal number of theoretical samples? It seems insufficient to compare only three different three different theoretical sample numbers (20, 40, and 80) .
Reviewer 2 Report
The article is interesting and well-written, but I think the analysis of the presented results should be expanded/detailed. The results are promising, but rather inconclusive, correlations arund 0.5 are low. However, the REF set of options for the forward rolling forecast seems to be the best, in comparison with the others (DOMAIN, PREDICTOR, SAMPLE and AMPLIFY). A suggestion to expand/detail the results is to include a table showing correlation of the observed rainfall for REF (and forward rolling forecast ) with the observed rainfall considering more anomaly percentages (not only +/- 25%, but also +/- 50%, +/- 10%, and +/- 5%.
Perhaps the main problem is the use of rain gauges to estimate rainfall. A rain gauge provides a very limited estimate of rainfall at a given point (the station). This may explain why results were not better. An alternative would be the use of the level of the Yangtze river at its outlet, since it is a kind of proxy estimate of the rainfall. Another alternative would be the use of weather radar at Nanjing to provide accumulated rainfall estimates over a period of time. Obviously, the problem of such alternative rainfall data is the availability of historical time series for many decades.
Major points:
Please consider the two following paragraphs:
(1st paragraph - abstract) "The multiple regression method is still an important tool for establishing precipitation forecast models with a lead time of one season"
(2nd paragrah - lines 204-206) "Assuming it was 1st March 1986, the July 1986 precipitation was predicted using the regression equation established by the observed samples from 1951 to 1985"
What does it mean "lead time of one season"? Reading the article, I understand that samples are yearly, not montly, i.e. there is one sample per year. To what month refers the yearly sample? The observed samples are from March for every year? If this is the case, the lead time would be one year plus one month.
The word "sensitivity" is used in two different contexts in many parts of the article, like (A) as a kind of sensisivity analysis (Table 2), or ike (B) lin Figure 5.
Minor corrections:
Please rewrite this phrase (lines 357-359) to make it clearer: "The prophase SST anomaly field 357 from 2001 is weaker than that from 1998, are not opposite to the sensitivity field in a few 358 regions (e.g., South China Sea)."
Please change computational efficiency to processing time, since the former expression refer to parallel efficiency. In line 228, REF was wrongly typed as ERF. Some names for the experiments may be misleading. For instance, I would suggest PCA (from Principal Component Analysis), instead of PREDICTOR, and NO-T-SAMPLE (not using theoretical samples) instead of SAMPLE, NO-AMPLIFY instead of AMPLIFY, G-DOMAIN instead of DOMAIN.
Reviewer 3 Report
Major comments
1) ABSTRACT: Is this linear correlation 0.5 or 1. What do you mean from 0.5?
2) Line 162: How many principle components are taken into consideration?
3) Line 171: Pearson correlation coefficient is valid for normal (Gaussian) probability distribution functions (PDF). Did the authors tested SST and precipitation data convenience with this PDF?
4) Line 179: What is the significance of 70 subsets or is it 70-year?
5) Line 188: Does "Cor" mean correlation or software? If a software please specify the features.
6) Lines 246-254: Only the number of peaks is given, but their comparison and percentages of closeness to the actual observations are not given.
7) Figure 3: Backward and forward predictions' comparison with observation data is possible visually that lead to subjective judgments. The authors can prepare a table to show the numerical statistical values (mean, standard deviation, skewness, correlation coefficient) for forecast and observation data so that the reader can have objective judgment.
8) Figure 4: The authors must prepare a table that shows numerically all the statistical parameters in addition to the numbers as mentioned in the text and Cor values in a single table numerically for observation, RegStep and RegLOO data values.
9) Lines 380-385: Ä°mportance of machine learning deep-learning and artificial neural network procedures cannot be denied for such predictions. They provide even far better results.
10) 397-399: The first two sentences are superfluous. Rational thinking and logical rules precedes theoretical and statistical knowledge. Even in probabilistic, stochastic and chaotic processes there is no fixed standard judgement.
11) Line 410: In the text, not only SST But also precipitation records are also modelled.
12) Lines 438-440: Not only statistical models but more extensively stochastic models (including statistical principles) are very flexible and still are in progress especially with the share of machine- and deep-learning procedures.
Reviewer 4 Report
The manuscript "Forecasting the July Precipitation over the middle-lower reaches of the Yangtze River with a flexible statistical model" is well-structured and has significant content in forecasting precipitation in monsoon season and is acceptable for publication.
Round 2
Reviewer 2 Report
OK, thanks for the corrections and explanations.
Reviewer 3 Report
The authors have responded satisfactorily to my comments.